# Regulation of Liver Glucose and Lipid Metabolism by Transcriptional Factors and Coactivators

**DOI:** 10.3390/life13020515

**Published:** 2023-02-13

**Authors:** Balamurugan Ramatchandirin, Alexia Pearah, Ling He

**Affiliations:** 1Department of Pediatrics, Johns Hopkins University School of Medicine, Baltimore, MD 21287, USA; 2Department of Pharmacology and Molecular Sciences, Johns Hopkins University School of Medicine, 600 N. Wolfe St, Baltimore, MD 21287, USA

**Keywords:** fatty liver, obesity, type 2 diabetes

## Abstract

The prevalence of nonalcoholic fatty liver disease (NAFLD) worldwide is on the rise and NAFLD is becoming the most common cause of chronic liver disease. In the USA, NAFLD affects over 30% of the population, with similar occurrence rates reported from Europe and Asia. This is due to the global increase in obesity and type 2 diabetes mellitus (T2DM) because patients with obesity and T2DM commonly have NAFLD, and patients with NAFLD are often obese and have T2DM with insulin resistance and dyslipidemia as well as hypertriglyceridemia. Excessive accumulation of triglycerides is a hallmark of NAFLD and NAFLD is now recognized as the liver disease component of metabolic syndrome. Liver glucose and lipid metabolisms are intertwined and carbon flux can be used to generate glucose or lipids; therefore, in this review we discuss the important transcription factors and coactivators that regulate glucose and lipid metabolism.

## 1. Introduction

Nonalcoholic fatty liver disease (NAFLD) is a major public health concern which affects one-fourth of the general population worldwide [1]. In the United States, NAFLD affects 30% of the population [2] and is expected to reach 100 million people in the United States by the year 2030 [3]. NAFLD is characterized by lipid accumulation in the cytoplasm of hepatocytes, which results from an imbalance in lipid acquisition, clearance, and export (i.e., fatty acid uptake, de novo lipogenesis and mitochondrial fatty acid oxidation, production of very low-density lipoproteins particles) [4]. NAFLD is predicted to become the second most common reason for liver transplantation in the United States [6,7].

Obesity, which has reached epidemic proportions worldwide, is associated with an increased risk of numerous metabolic abnormalities, such as NAFLD and T2DM [1,2,3]. Diabetes, T2DM in particular, is a major health burden because of its chronic nature, growing speed, medical cost, and its impact on adults and adolescents [4]. In the United States, diabetic prevalence is estimated to be totaled at approximately 70 million by the year 2050; furthermore, currently 422 million adults globally are living with diabetes and this number is expected to increase to 629 million according to the World Health Organization (WHO) [5] (www.cdc.gov/diabetes/data/statistics-report (accessed on 6 March 2022). In children and adolescents, the prevalence of both type 1 diabetes mellitus (T1D) and T2DM has also increased [6]. Diabetes is associated with increased risk of chronic liver disease, cardiovascular disease, stroke, infections, chronic kidney disease, cancer, diabetic neuropathy, and blindness [7]. The link between diabetes and liver damage, such as fatty liver, cirrhosis, and hepatocellular carcinoma, is well documented [8]. Low-grade inflammation plays a critical pathophysiological role in diabetes [9] and patients with diabetes mellitus have an increased risk of developing infections as well as sepsis; this constitutes 20.1–22.7% of all sepsis patients [10].

T2DM, accounting for more than 90% of diabetes [4], is a severe and complex disease. Because of insulin resistance and insufficient secretion of insulin from pancreatic β cells, increased glucose production in the liver along with reduced glucose utilization in insulin-sensitive tissues, such as muscle and adipose tissues, lead to hyperglycemia in these patients [11,12,13]. Abnormal glucose metabolism impacts the major metabolic pathways, such as carbohydrates and lipids within different tissues and organs [14]. In mammals, the liver is the largest metabolic organ responsible for maintaining long-term energy needs in the body, which includes liver glucose metabolism and lipid metabolism that are tightly intertwined, and liver metabolism that is largely regulated by various transcriptional factors and coactivators. In this review, we will discuss the current understanding of the importance of transcriptional factors and coactivators in the regulation of liver glucose and lipid metabolism.

## 2. Regulation of Liver Metabolism by Transcription Factors

Evolution is tightly linked to phenotypic variations of organisms by gene expression. A change in the functional product of each gene is altered primarily by transcription factors. Transcription factors can bind to the promoter or enhancer regions of specific genes through their DNA-binding domains [15]. As a result, transcription factors regulate gene transcription, protein synthesis, and cellular function. Glucose is one of the principal energy fuels for mammalian cells and takes a central position in metabolism. In some mammalian tissues and cells, such as neurons, erythrocytes, and renal medulla, glucose is the sole energy source. Low blood glucose levels (hypoglycemia) can cause serious damage to these tissues or cells [16,17]. On the other hand, high blood glucose levels (hyperglycemia) can cause serious adverse effects, which are clearly demonstrated in diabetic patients [18,19]. Therefore, maintenance of blood glucose levels within a narrow range (70~110 mg/dL) is critical for protection of organisms against hypoglycemia during fasting and postprandial hyperglycemia. The liver is one of the main organs that contributes to the regulation of blood glucose levels by releasing the glucose into circulation when blood glucose levels are low and stores along skeletal muscles excess of glucose as glycogen in postprandial states. In mammals, the liver is the central organ for fatty acid metabolism and a key player in glucose metabolism. The liver acts as a crossroad, which metabolically connects various organs, especially skeletal muscle, and adipose tissues, in the endeavor to maintain the long-term energy supply to the body. The regulation of liver glucose metabolism and lipid metabolism is tightly controlled by dietary, hormonal, and neural signals through the activation of various transcriptional factors and coactivators, which will be discussed.

### 2.1. Cyclic AMP Response-Element Binding Protein (CREB)

CREB, a transcription factor, contains a basic leucine zipper motif and a DNA-binding basic region. CREB binds to the cAMP response element (CRE, 5′-TGACGTCA-3′) [20] in the promoter region of rate-limiting gluconeogenic gene, such as *G6pc* and *Pck1,* to upregulate gluconeogenesis in the liver. In the fast state, elevated blood glucagon activates the cAMP-PKA pathway and the phosphorylation of CREB at S133 by PKA, leading to the recruitment of CBP/P300 to CREB (Figure 1) [21,22]. Activated PKA also mediates the dephosphorylation of CRTC2 [21,22]. Phosphorylation of CRTC2 by SIK2 excludes CRTC2 from the nucleus; however, activated PKA can phosphorylate SIK2 at Ser587 and suppress SIK2 activity, therefore, negating its inhibition on CRTC2 [23], thus preventing CRTC2 degradation. Dephosphorylation of CBP leads to its association with CREB [24]. These events result in the formation of the CREB-CBP/P300-CRTC2 coactivator complex to promote gluconeogenic gene expression and hepatic glucose production (Figure 1). Interestingly, knockdown of the liver CREB by antisense oligonucleotide not only alleviated the hyperglycemia, but also reduced liver lipogenesis [25], suggesting that CREB can modulate both glucose and lipid metabolism.

### 2.2. FOXO1

FOXO1 transcription factor belongs to the large Forkhead family of proteins containing a highly conserved DNA-binding domain termed the “Forkhead box”. There are four members (FOXO1, FOXO3, FOXO4, and FOXO6) in the FOXO subgroup [26]. Among these proteins, FOXO3 is highly expressed in the brain and FOXO6 is predominantly expressed in the developing brain. FOXO1 is abundant in adipose tissues and liver, and FOXO4 is abundant in muscle tissues [27]. The Forkhead box, a 110 a.a. region located in the N-terminal part of FOXO protein, can bind to DNA containing two of the consensus recognition DNA sequences: DAF-16 family member-binding element (5′-GTAA(C/T)A-3′) [28] or insulin-responsive element (5′-(C/A)(A/C)AAA(C/T)AA-3′). All of the members in the FOXO family can recognize the core DNA sequence (5′-(A/C)AA(C/T)A-3′) [29].

FOXO1 is a major target of insulin signaling and is the direct substrate of protein kinase B (AKT) in response to the stimulation of insulin or growth factors. Phosphorylation of FOXO can alter their subcellular location, DNA binding affinity, and transcriptional activity [30]. Phosphorylation of FOXO1 at S256 by AKT generates a negative charge in the DNA-binding domain, thus hindering DNA binding [31]. The phosphorylation events in FOXO1 can increase its association with 14-3-3 proteins, leading to the translocation of FOXO1 from nucleus to cytoplasm [32]. Therefore, FOXO1 is rendered to inactive and degradation by virtue of its subcellular compartmentalization and diminished DNA binding (Figure 2). In the liver, FOXO1 binds to the promoter region of rate-limiting gluconeogenic genes *Pck1* and *G6pc* to upregulate liver glucose production [33,34,35]. Recent studies from Dr. Guo’s group showed that phosphorylation of FOXO1 at S276 by PKA can retain FOXO1 in the nucleus and increases its stability (Figure 1) [35]. Interestingly, FOXO1 and IRS2 can reciprocally increase each other’s stability and generate a regulatory circuit [36]. FOXO1 can affect the VLDL production through regulation of the expression of the microsomal triglyceride transfer protein (MTP) or ApoC3 [37,38].

### 2.3. Carbohydrate Response Element Binding Protein (ChREBP)

ChREBP is an important transcriptional activator of hepatic lipogenesis and fructose metabolism [39,40]. ChREBP binds to a highly conserved carbohydrate response element, C(A/G)(C/T)G(T/C/G)Gnnnnn(C/A)(C/G/A)(C/T/G)G(T/A/G)G, on the promoter of target genes [41]. ChREBP is activated by increased glucose influx and drives the gene expression of the rate-limiting glycolytic pyruvate kinase in liver, and acts in synergy with SREBP-1 (see below) to drive expression of genes involved in de novo fatty acid synthesis (lipogenesis) and esterification, such as ACC, FASN, SCD1, and Elovl6 (Figure 3) [41,42]. Therefore, in the feeding state, excessive blood glucose can be split into pyruvate and channeled into lipogenesis in the liver. Additionally, in the feeding state, glucose will be converted to xylulose 5-phosphate through the pentose phosphate pathway to glucose 6-phosphate (Figure 3). Activation of PP2A by xylulose 5-phosphate, and subsequently, the dephosphorylation of ChREBP by PP2A [41], together with the binding of glucose 6-phosphate to ChREBP render its translocation to the nucleus [43]. ChREBP plays a role in regulating fructose metabolism because loss of liver ChREBP reduces the expression of fructokinase and triose kinase, leading to fructose intolerance [40]. In the fasting state, activated PKA by elevated blood glucagon phosphorylate ChREBP at S196 and T666 leads to the disassociation of ChREBP from the promoters of target genes and binding to 14-3-3, which excludes ChREBP from the nucleus and retains it in the cytoplasm [41,44]. Since ChREBP can activate *G6pc* expression, ingested fructose can be converted to glucose in the liver and be released into circulation to exacerbate hyperglycemia [45]. Conversely, the gene expression levels of ChREBP are positively correlated with hepatosteatosis and negatively related to liver glycogen contents and insulin resistance.

### 2.4. Sterol Regulatory Element-Binding Protein-1 (SREBP-1)

SREBPs, transcription factors and the master regulators of cellular lipid metabolism and homeostasis, are involved in various physiological and pathological processes of lipid synthesis [46]. There are two SREBP genes, SREBP-1 and SREBP-2, that give rise to three SREBP proteins: SREBP-1a, SREBP-1c and SREBP-2. SREBP-1 is mainly involved in fatty acid synthesis and SREBP-2 in cholesterol synthesis. SREBP-1c and SREBP-2 are the predominant isoforms expressed in the liver. The regulation of these gene expressions is in a feed-forward manner because their promoters harbor the sterol regulatory element (SRE, 5′-ATCACNCCAC-3′) [47]. To activate the transcription of target genes, SREBPs must be transported to the Golgi complex, where they are cleaved by two functionally distinct proteases, namely, Site-1 protease (S1P, also known as MBTPS1) and Site-2 protease (S2P, also known as MBTPS2). Then, the transcriptionally active fragment of SREBP is released into the cytosol and then migrates into the nucleus. This activation by proteolytic processing is tightly regulated by the interaction with ER membrane proteins, SREBP cleavage-activating protein (SCAP), and insulin-induced gene (Insig) [48]. Under conditions of deprived sterols, SCAP escorts SREBPs from the ER to the Golgi complex through incorporation into COPII-coated vesicles. Subsequently, SREBPs are cleaved by S1P and S2P within the Golgi. With increased cholesterol content (high sterols), Insig binds to SCAP to retain SCAP/SREBPs on the ER membrane as an inactive precursor [49].

SREBP-1c preferentially upregulates gene expressions that are related to fatty acids and triglyceride synthesis, such as ACC, FASN, SCD1, and GPGT, while SREBP-2 mainly controls gene expressions required for cholesterol synthesis and uptake, such as HMGS, HMGR, and LDL receptor [46]. Insulin or feeding can drastically increase the mRNA levels of SREBP-1c [50,51], through the mTOR, and requires LXR and C/EBPβ (Figure 3) [52]. Insulin also regulates the expression of Insig-1 and Insig-2, two important inhibitors for the proteolytic activation of SREBPs [53,54]. Since overexpression of either SREBP-1c or SREBP-1a in mouse liver markedly increased the expression of lipogenic genes and causes massive hepatosteatosis [55], hyperinsulinemia in obesity or early stage of T2DM would stimulate the expression of SREBP-1, thus leading to the development of steatosis.

### 2.5. X-Box Binding Protein 1 (XBP1)

Endoplasmic reticulum (ER) stress has emerged as a key player in the progression of insulin resistance and the promotion of lipid accumulation in the liver since it can transduce some effects of lipid metabolites and cytokines into the activation of stress kinases [56]. ER stress leads to the cellular response termed the unfolded protein response (UPR) through the activation of three canonical pathways: IRE1-XBP1s, PERK-eIF2, and ATF6 [57,58]. Solid evidence has demonstrated the elevation of the UPR in the liver of patients with diabetes, obesity, and NAFLD, whereas administration of the chemical chaperones 4-phenyl butyric acid (PBA) or taurine-conjugated ursodeoxycholic acid (TUDCA) to *ob/ob* mice normalized blood glucose levels, improved insulin sensitivity, and reduced hepatic steatosis [59,60]. ER stress is able to induce the entire lipogenic program, and the attenuation of ER stress in *ob/ob* mice which decreases SREBP1c activation and triglyceride contents in the liver [61]. In particular, the IRE1-XBP1 pathway is critical for lipogenesis [62,63].

XBP is a bZIP member of the CREB/ATF family of transcription factors. This gene consists of six exons and encodes 261 amino acids in the normal condition. However, the accumulation of unfolded proteins in the ER leads to the activation of IRE1α, and activated IRE1α excises a 26-nucleotide intron of the XBP1 mRNA and shifts the coding reading frame, resulting in the expression of a more stable and active form known as XBP1s (for the spliced form) with 276 amino acids [64,65]. XBP1s is an important transcription factor that drives SREBP1 gene (Srebf1) expression [66]. Coactivator P300 can be induced by XBP1s, thus increasing glucose production in the liver (see below) [67,68]. Our recent study revealed that the activation of IRE1-XBP1 signaling by insulin-activated AKT increases XBP1s to stimulate liver lipogenesis in the fed state (Figure 3), while in the fasted state augmentation of XBP1u (the unspliced form) upregulates liver gluconeogenesis to maintain the euglycemia [69]. Thus, XBP1s can affect both glucose and lipid metabolism.

### 2.6. Nuclear Receptors

Nuclear receptors (NRs) are ligand-regulated transcriptional factors involved in a broad spectrum of biological processes [70]. No ligand-binding orphan receptors were also included in this superfamily, and currently, this superfamily comprises of 48 members. NRs play several roles in the regulation of physiological processes of metabolism, development, cell proliferation, immune response, reproduction, and pathological processes such as metabolic syndromes and cancer, as well as neurological disorders. Members in this family share similar protein homology, a transactivation domain at the N-terminal, a DNA-binding domain followed by a hinge domain, and a ligand-binding domain at the C-terminal [71]. The binding of a ligand to the ligand-binding domain transforms the NRs into transcriptionally competent factors. Based on structural and phylogenetic analysis, NRs are divided into six distinct groups. The dysregulation of NRs contributes to the development of several diseases, including diabetes, cancer, etc. Several NRs can play important roles in regulating liver glucose and lipid metabolism which are discussed below.

#### 2.6.1. Peroxisome Proliferator Activated Receptors (PPARs)

Discovered 30 years ago, PPARs (consisting of three isotypes: PPARα, β/δ, and γ) are important metabolic regulators of systemic energy homeostasis. They are ligand-inducible transcriptional factors that belong to the nuclear receptor superfamily. However, each of them contains different functions and expression abundance within the targeting tissues [72]. All PPARs, like other NRs, have an N-terminal activation domain, a central DNA-binding domain, a cofactor-binding flexible hinge region, and a ligand-binding domain at the C-terminal. Ligand binding, heterodimerization with RXR, and interaction with coactivators lead to the full activation of transcriptional activity of genes containing PPAR response element (PPRE, 5′-AGGTCANAGGTCA-3′) and each PPAR member is preferentially bound by distinct ligands [73]. PPARs can be activated by natural lipid-derived metabolites, dietary lipids, and synthetic compounds, including fibrates and TZD, to augment target gene expression in the liver, muscle, and adipose tissues. 

PPARα is highly expressed in the metabolic tissues, such as liver and skeletal muscle, with high mitochondrial fatty acid oxidation. The expression of PPARα is regulated by insulin, leptin, growth hormones, glucocorticoids, and stress. Fasting increases fatty acid oxidation by induction of PPARα to provide ATP and GTP for the energy-demanding gluconeogenesis process in the liver. Leptin can upregulate PPARα expression through changes of fatty acids flux [74]. In the presence of PGC-1α, adiponectin also upregulates PPARα-responsive gene expression [75]. However, both insulin and growth hormone downregulate PPARα expression [76,77]. PPARα target genes can regulate lipid and glucose metabolism and inflammatory responses [78]. PPARα functions primarily in the liver and controls directly or indirectly the gene expressions of FATP, CD36, FABP, CPT1α and 2, LCAD, VLCAD, ECI1, and ACOX to regulate fatty acids uptake, binding, transport, synthesis, oxidation, storage, and ketogenesis [79,80]. Liver-specific PPARα knockout mice exerted increased liver lipid accumulation and hypercholesterolemia [81]. In the fed state, PPARα can control lipogenesis and allows the use of nutrients to generate fatty acids [82] because activation of PPARα increases the levels of mature nuclear form SREBP-1c via proteolytic cleavage [83].

PPARβ/δ is expressed ubiquitously as well as in most metabolically active tissues and plays a role in fatty acid oxidation and glucose metabolism. Activation of PPARβ/δ ameliorates hyperglycemia by reducing liver glucose production and increasing lipogenesis in the liver, and HFD-fed PPARβ/δ null mice exhibited increased VLDL production [84]. PPARγ consists of γ1, γ2, and γ4 isoforms in humans and is abundant in adipose tissues, but γ2 can be induced in the liver and muscle tissues [85,86]. PPARγ has an essential role in regulating adipocyte differentiation, lipid storage, insulin sensitivity, and glucose metabolism [87]. Activation of PPARγ by TZDs improves hyperglycemia and insulin sensitivity in patients with T2DM [88]. Knockout of liver PPARγ reduced liver steatosis, but elevated blood triglycerides and free fatty acids in addition to exacerbating hyperglycemia and insulin resistance [89].

#### 2.6.2. Liver X Receptors (LXR)

LXRs, consisting of two isoforms (LXRα/β), are the transcriptional factors of the NR superfamily and play an important role in the control of cholesterol and lipid metabolism [90]. LXRα is highly expressed in the metabolic tissues—liver, intestine, adipose tissues, and macrophages—while LXRβ is expressed ubiquitously [91]. LXRs heterodimerize with retinoid X receptor (RXR) and bind to target gene promoters on LXR-responsive-element (LXRE). The canonical LXRE is composed of the repeated sequence AGGTCA, separated by four nucleotides [92]. A number of cholesterol derivatives, such as oxysterols and desmosterol, and synthetic ligands, can function as LXR activators [90,93]. Without the ligand binding, the LXR/RXR complex binds to corepressors: nuclear receptor corepressor (NCoR) and the silencing mediator of retinoic acid and thyroid hormone receptor (SMRT) to inhibit target gene expression. When either LXRs or RXR are activated by their ligands, the corepressors will be replaced by coactivators P300 and nuclear receptor coactivator 1, leading to the expression of target genes related to lipid metabolism [94]. The LXRα knockout mice had massive accumulation of cholesterol in the liver when fed a high-cholesterol diet [95]. Activation of LXRs increase bile acid synthesis and biliary cholesterol excretion through upregulation gene expression of *Cyp7a1* and *Abcg5/8*, respectively [95,96]. Activation of LXRs stimulates transport from peripheral tissues to the liver for bile acid synthesis and excretion, thus decreasing the cellular cholesterol levels accomplished through the upregulation of the expression of the ATP bonding cassette (ABC) A1/G1 and reducing the low-density lipoprotein receptor (LDLR) [97]. Activation of LXRs also increased liver triglyceride levels through increasing de novo lipogenesis by upregulating the expression of SREBP1c and ChREBP [98,99].

#### 2.6.3. Farnesoid X Receptors

Farnesoid X receptor (FXR) is a nuclear receptor and a ligand-activated transcriptional factor involved in the regulation of bile acid synthesis by controlling bile acid synthesis, transport, and metabolism, the enterohepatic cycle, glucose and lipid homeostasis, oxidative stress, and inflammation [100]. Four FXR isoforms are generated through an alternative promoter and RNA splicing. FXR is mainly expressed in the liver, intestines, and kidneys. In the liver, FXR modulates bile acid synthesis via inhibition of CYP7A1 [101]. FXR stimulates bile acid secretion by increasing the bile acid export pump and multidrug related protein, meanwhile, FXR reduces bile acid reabsorption. Activation of FXR protects against liver accumulation of bile acids; consequently, FXR knockout mice exhibit significantly increased bile acid levels in the liver, suggesting that FXR primarily regulates bile acid homeostasis and is a therapeutic target of cholestatic liver diseases [102]. Furthermore, FXR activation leads to reduction of liver triglyceride levels through upregulation of carboxylesterase 1 and PPARα as well as downregulation of SREBP1c [103,104]. FXR knockout mice exhibit increased blood glucose levels and insulin intolerance; however, activation of FXR reduces liver gluconeogenesis by inhibiting the expression of *Pck1* and *G6pc* [103].

#### 2.6.4. Hepatocyte Nuclear Factor 4 (HNF4)

HNF4 is a transcriptional factor belonging to the orphan nuclear receptors in the NR superfamily and its expression and activities are restricted to the liver and gastrointestinal tract while containing two isoforms (HNF4α and HNF4γ) [105,106]. HNF4 proteins have an additional repressor domain with an inhibitory function [107]. Interestingly, fatty acids could present constitutively in the ligand-binding domain, making HNF4 different from other members in the NR superfamily [108]. HNF4 has critical roles in liver development and regulation of liver functions through controlling the expression of liver-specific genes related to glycolysis, gluconeogenesis, fatty acid metabolism, urea production, bile acid synthesis, apolipoprotein synthesis, and drug metabolism [109]. Inactivation mutations in HNF4α result in insulin secretion defects [110]. Liver-specific HNF4α-null mice exhibited increased fat accumulation in the liver, drastic reduction of liver gluconeogenesis, along with reduced blood cholesterol and triglyceride levels [111]. Post-translational modification of HNF4 can modulate its transcriptional activity [112,113].

## 3. Regulation of Liver Metabolism by Coactivators and Corepressors

The genetic material of DNA is wrapped around core histones that form nucleosomes, the fundamental organizing unit, within the chromatin of all eukaryotic genomes [114]. The efficient packaging of the genetic material in the chromatin forms a physical barrier to RNA polymerase and transcription factors for preventing unnecessary gene transcription. To initiate gene transcription, histones undergo post-translational modifications (PTM) and changes to their chemical properties to unwrap chromatin. The core histones (H2A, H2B, H3, and H4) are rich in positively charged Lys residues, which bind to the negatively charged DNA backbone. However, acetylation masks the positive charges leading to the condensed chromatin being loosened and the chromatin being opened in order to be transcriptionally accessible to transcriptional factors and coactivators as well as the formation of the transcription apparatus [115,116]. Acetylation is a reversible process in which histone acetyltransferases (HATs) add an acetyl group to the lysine residue of the target protein, whereby, histone deacetylases (HDACs) remove the acetyl group. HATs and HDACs are major histone family proteins which regulate the post translational modifications process [117].

### 3.1. Regulation of Liver Metabolism of Coactivators CBP and P300

Coactivators can interact with other transcriptional factors and nuclear receptors to increase the gene transcription rate and are required for effective transcription of many eukaryotic genes through modification of constraint chromatin and activator recruitment of the transcriptional apparatus. To date, acetylation is the best understood and well-studied histone modification and the status of histone acetylation is closely related to transcriptional activation and silencing. HAT transfers an acetyl group to the ξ-amino groups of the lysine side chain within a histone or target protein, while the acetyl group can be removed by HDACs. HAT is classified by three different types based on their sequence homology, structure, and functional similarity.

In humans, there are 30 known HATs which are grouped into five families: GCN5- related N-acetyltransferase, MYST, CBP/P300, the general transcription factor HATs, and the nuclear hormone-related (SRC/NCoA) HATs [118,119]. GCN5 can regulate glucose and iron metabolism [120,121]. MYST histone acetyl transferases have been implicated in uncontrolled cell growth and malignancy in several human cancer types [122]; however, their role in regulating metabolism is not well documented. CBP/P300 contains three cysteine–histidine-rich domains, bromodomain, and an acetyl transferases domain that are involved in several signal transduction pathways to regulate metabolism and energy homeostasis [123,124]. CBP/P300 can act as histone acetyltransferases as well as transcriptional coactivators. The acetyltransferase activity of CBP/P300 transfers the acetyl group to lysine residues in histones, causing chromatin remodeling, and these proteins can bind to the transcriptional start sites and function as a coactivator with transcriptional factors to initiate target gene expression. CBP and P300 are ubiquitously expressed [125]. They are closely related proteins sharing numerous near identical regions and a similar structure including four cysteine–histidine-rich regions (CH1-3), the KIX domain (CREB-binding site), the bromodomain, the HAT domain, and the steroid receptor coactivator-1 interaction domain (SID) [126].

To protect organisms against hypoglycemia during fasting and postprandial hyperglycemia by the opposing actions of insulin signaling and glucagon signaling pathways, blood glucose levels are maintained within a narrow, defined range (70–110 mg/dL, fasting). In the fasted state, glucagon stimulates liver glucose production through the cAMP-PKA signaling pathway and phosphorylation of CREB at S133 by PKA (Figure 1) [127]. Phosphorylation of CREB at Ser133 leads to the recruitment CBP, P300, and CRTC2 to CREB and the formation of the CREB coactivator complex on the promoters of the rate-limiting gluconeogenic genes, *Pck1* and *G6pc,* to stimulate gene expression and glucose production in the liver [21,24,128]. P300 can acetylate CRTC2 at K628 to prevent CRTC2 nuclear exclusion and degradation [129]. Furthermore, FOXO1 upregulates gluconeogenesis through activation of *Pck1* and *G6pc* gene expression in the liver [130,131]. Our study showed that P300 stimulates *Foxo1* gene expression in the fasted state through binding to the tandem cAMP-response element sites in the proximal promoter region of the *Foxo1* gene [132]. Since FOXO1 binds to insulin responsive sequence (IRS) on the gluconeogenic gene, the coordinated effects of P300 and FOXO1 can fully activate the gluconeogenic program (Figure 1).

In the fed and postprandial states, elevated insulin levels will reduce blood glucose levels, and the gluconeogenic engine—the CREB-CBP/P300 complex—needs to be turned off and glucose will be used to generate glycogen and stored in the liver. Interestingly, insulin can also stimulate the phosphorylation of CREB at S133 [21], suggesting that CREB is constitutively phosphorylated at S133. CBP can be phosphorylated at S436 by atypical PKC*ι*/*λ* (Figure 2) [22], leading to its dissociation from the CREB coactivator complex. However, P300 lacks the corresponding S436 phosphorylation site found in CBP, therefore, P300 constitutively binds to CREB to maintain the basal gluconeogenesis in the fed and postprandial states [24]. Physiologic concentration of glucose could not effectively promote glycogen synthesis when glucose was the sole substrate and efficient glycogen synthesis occurred when gluconeogenic precursors were added [133]. It has been proven that the gluconeogenic pathway contributes for 50–70% of newly synthesized glycogen during the postprandial state [134,135,136]. To mimic the phosphorylation site found in CBP, a phosphorylation-competent P300G422S knock-in mouse model was generated. These P300 knock-in mice had significantly decreased liver glycogen content and produced significantly less glycogen in a tracer incorporation assay in the postprandial state, indicating an important and unique role of P300 in glycogen synthesis through maintaining basal gluconeogenesis [137]. Moreover, both P300 and CBP can acetylate other proteins such as FXR, SREBP-1, and FOXO1 to modulate gene expression related to glucose and lipid metabolism (see below section) [138,139,140].

### 3.2. Regulation of Liver Metabolism of CREB-Regulated Transcription Coactivator 2 (CRTC2)

CRTC proteins, also known as transducers of regulated CREB activity, regulate the metabolic flux by controlling the important enzymes within the metabolic process. CRTC proteins have CREB-binding domain (amino terminus), transactivation domain (carboxy terminus), and regulatory domain (central part) which bind to the CREB and other bZIP (basic leucine zipper) proteins, basic transcriptional machineries, and enzymes [20]. Among the CRTC isoforms, CRTC2 can regulate glucose and lipid metabolism in a tissue-specific manner, including the liver, pancreas, the small intestines, and adipose tissues [141]. During fasting conditions, CRTC2 increases the gluconeogenic flux in the liver by activating *Pck1* and *G6pc* genes [21]. Activated AKT by insulin through the PI3K-AKT pathway can activate SIK2, leading to the phosphorylation of CRTC2 at Ser171, and subsequently, the phosphorylated CRTC2 is excluded from the nucleus and degraded in the cytoplasm (Figure 2) [21]. However, activated PKA can drive CRTC2 translocation into the nucleus through inhibition of SIK2 [23].

### 3.3. Regulation of Liver Metabolism of PGC-1α

Peroxisome proliferator activated receptor gamma coactivator-1 alpha (PGC-1α) is a master transcriptional coactivator for mitochondrial biogenesis in brown adipose tissues [142]. This coactivator also regulates the transcription of genes related to gluconeogenesis, glucose transport, glycogenolysis, fatty acid oxidation, energy homeostasis, and mitochondrial activity in the liver [143,144,145]. In the fasted state, activation of cAMP-PKA by glucagon leads to the phosphorylation of CREB at S133 and recruitment of CBP to the promoter of *PGC-1α* to stimulate the *PGC-1α* gene expression, subsequently, PGC-1α coactivates HNF4α to upregulate the rate-limiting gluconeogenic gene expression of *Pck1* and *G6pc* (Figure 1) [143,146]. Ιnterestingly, PGC-1a can affect insulin signaling and glucose metabolism by regulating the ratio of IRS1 and IRS2 in the liver [147].

PGC-1α has an important role in the regulation of lipid metabolism in the liver as well [148]. Mitochondrial dysfunction results in decreased fatty acid oxidation and is implicated in the development of hepatic steatosis. Overexpression of PGC-1α in hepatocytes elevates mitochondrial content and function, leading to an increase in fatty acid oxidation and reduction of triglyceride levels and VLDL synthesis and secretion [149]. On the other hand, heterozygous liver-specific PGC-1α knockout mice exhibited decreased expression of genes related to mitochondrial β oxidation and accumulation of triglycerides in the liver [144]. Furthermore, mice fed on a choline-deficient diet increased fat accumulation in the liver in part through impaired PGC-1α activity [150].

### 3.4. Regulation of Liver Metabolism of Corepressors

Corepressors comprise multiple proteins including DNA-binding proteins, histone methyltransferases, HDACs, Silencing mediator of retinoic acid and thyroid hormone receptor (SMRT), Nuclear receptor corepressor (NCoR), G-protein sippressor 2 (GPS2), transducing β-like protein (TBL1), and TBL-related 1 (TBLR1). Corepressors function as transcriptional silencers through interaction with many transcriptional factors, resulting in inactivation of a variety of chromatin remodeling enzymes to alter metabolism [151,152,153,154]. NCoR/SMRT/HDAC3 have a large NR-binding surface that interacts with multiple NR targets, thereby to regulate the liver metabolism [155]. Notably, HDAC3 can interact with PPARγ to impair insulin sensitivity in obese models and NCoR/HDAC3 can alter lipogenic enzymes through MeCP2 (Metyl-CpG-binding protein 2) [156]. In line with these studies, depletion of NCoR1/SMRT leads to metabolic abnormalities, including hypoglycemia, hypothermia, and weight loss [156].

## 4. Coordinated Regulation of Glucose and Lipid Metabolism by Transcriptional Factors and Coactivators in the Liver

### 4.1. Coordinated Regulation of Glucose Metabolism in the Liver

Blood glucose levels are tightly modulated by the insulin and glucagon signaling pathways. In the fed and postprandial states, elevated blood glucose levels promptly stimulate the secretion of insulin from pancreatic 𝛽 cells, leading to increased glucose uptake and utilization in insulin-sensitive tissues such as muscle and adipose tissue, and suppression of liver glucose production. In the fasting state, glucagon, secreted from the 𝛼 cells of the pancreas, stimulates liver gluconeogenesis through upregulation of the rate-limiting gluconeogenic genes *Pck1* and *G6pc*. These two gluconeogenic genes are also controlled at the transcriptional level by insulin (discussed below).

In the fasted state, activation of cAMP-PKA signaling by elevated blood glucagon will lead to increased CREB phosphorylation at S133 and the formation of CREB-CBP/P300 complex to drive the expression of rate-limiting gluconeogenic expression (Figure 1). Augmentation of the expression of *PGC-1α* gene expression by CREB phosphorylation at S133 and recruitment of CBP leads to upregulation of the rate-limiting gluconeogenic gene expression of *Pck1* and *G6pc* via HNF4α [143,146] (Figure 1). The salt-inducible kinases 2 (SIK2) play a role in modulating CRTC2 activity through phosphorylation of CRTC2 at S171 to exclude CRTC2 from the nucleus [21,23]. However, activated PKA can phosphorylate SIK2 at Ser587 to inhibit SIK2 activity and negate its inhibition on CRTC2, leading to CRTC2 translocation into the nucleus. Activation of cAMP-PKA signaling also stimulates the mRNA levels of the *FOXO1* gene driven by CREB and P300, which bind to tandem CRE sites in the proximal promoter region of the FOXO1 gene [132] (Figure 1). In addition, PKA-mediated phosphorylation of FOXO1 at S276 increases FOXO1′s stability and nuclear localization (Figure 1) [35]. The finely tuned and coordinated regulation of gluconeogenic gene by these transcriptional factors and coactivators will maintain the glucose levels within the normal range and provide enough glucose for tissues and cells that depend on glucose as the sole energy source.

In the fed state, elevated blood glucose levels promptly stimulate the secretion of insulin from pancreatic β cells, leading to the suppression of liver glucose production. Insulin can inhibit liver glucose production through several mechanisms to maintain blood glucose levels. To turn off the gluconeogenic engine, the CREB coactivator complex, activated aPKC*ι*/*λ* (atypical protein kinase C) by insulin, through the PI3K-PDK1 pathway, can directly phosphorylate CBP at S436 resulting in the disassembly of CREB coactivator complex (Figure 2) [22]. As mentioned above, insulin-stimulated phosphorylation of CRTC2 at Ser171 leads to nuclear exclusion and degradation (Figure 2). In addition, activated AKT mediates the phosphorylation of FOXO1, and then triggers the export of FOXO1 from the nucleus to the cytoplasm and promotes its ubiquitinylation and degradation (Figure 3) [29].

### 4.2. Coordinated Regulation of Lipid Metabolism in the Liver

In the fed state, elevated blood glucose and amino acids provide the substrate acetyl-CoA for lipogenesis, and also elevated blood glucose levels promptly stimulate the secretion of insulin from pancreatic β cells. In the liver, activation of the PI3K-AKT signaling by insulin results in increased expression of SREBP1c through mTOCR1 (Figure 3) [157]. Activated AKT can directly phosphorylate IRE1 at S724 to mediate the splicing of XBP1 mRNA and the generation of spliced form XBP1s; subsequently, XBP1s stimulates SREBP1c expression (Figure 3) [69]. Elevated blood insulin levels can increase the glucokinase activity to augment the generation of glucose 6-phosphate, then, xylulose 5-phosphate via the pentose-phosphate pathway. Xylulose 5-phosphate can promote ChREBP1 nuclear entry and lipogenesis (Figure 3) [158]. Furthermore, activated AKT mediates phosphorylation of FOXO1 and nuclear exclusion leads to the decreased expression of microsomal triglyceride transfer protein (MTP), limiting the export of triglyceride from the liver and favoring triglyceride accumulation in the liver [69].

## 5. Excessive Liver Glucose Production in T2DM and Obesity

In patients with T2DM and obesity, insulin resistance is the hallmark [1,56]. Because of intra-islet regulation and insulin resistance that lead to failed suppression of glucagon secretion from the pancreatic α cells, these patients have hyperglucagonemia and/or an increased ratio of glucagon/insulin [159,160,161,162]. This results in excessive liver glucose production and decreased glucose utilization in extrahepatic tissues (muscle and adipose tissues) causing fasted hyperglycemia. These patients have decreased liver glycogen contents [163,164]. The inappropriate high rates of liver glucose production, even in the compensatory stage with hyperinsulinemia, are through gluconeogenesis and glycogenolysis; however, the main contributor of inappropriate liver glucose production is elevated gluconeogenesis [165,166]. The availability of the gluconeogenic precursors is essential for the gluconeogenic capacity of the liver. Skeletal muscle is crucial for glucose utilization and is responsible for over 80% of glucose clearance [167,168]. However, up to 15% of glucose taken by the muscle is released as lactate, pyruvate, and alanine into circulation [169] and these metabolites will be used as gluconeogenic precursors by the liver. Importantly, due to insulin resistance and elevated ratio of glucagon/insulin in obesity and T2DM [159,160,161,162], lipolysis would be increased in adipose tissue and a glycerol flux to the liver would occur, glycerol will be used as a preferred gluconeogenic precursor (Figure 4) [170].

LPS leakage from the intestine into circulation and endoplasmic reticulum (ER) stress play important roles in the development of insulin resistance in obesity and T2D [56,171,172,173]. Our study showed that endotoxemia can induce P300 via XBP1s in the ER stress pathway in the liver [67]. Increased P300 will bind to the promoter of the *Foxo1* gene and upregulate the expression of the *Foxo1* gene (Figure 4) [132]. In addition, elevated P300 can bind to CREB to drive the gene expression of *Pck1* and *G6pc.* These data reveal the significance of changes in the composition of gut microbiota and the increase in intestinal permeability and LPS leakage, and their initiated low-grade inflammation in the development of hyperglycemia in T2DM and obesity. Because of insulin resistance and impaired insulin signaling [68], patients with obesity and T2DM have low AKT activity and impaired phosphorylation of FOXO1and Crtc2 by AKT, leading to increased FOXO1 and Crtc2 protein levels that drive further gluconeogenic gene expressions (Figure 4).

## 6. Abnormal Lipid Metabolism in the Liver of T2DM and Obesity

In obesity and T2DM, insulin resistance results in elevated blood glucose (hyperglycemia) and hyperglycemia can stimulate glucose uptake by the liver through mass action even with impaired activity of glucokinase (hexokinase IV) [174,175]. However, glucose from circulation would not be used to make glycogen due to the inappropriate activation of glycogen phosphorylase and inhibition of glycogen synthase by increased blood glucagon or glucagon/insulin ratio, leading to low glycogen contents in the liver [163,164]. Glucose carbon can therefore be used to make fatty acids, providing a nutrient flux for de novo lipogenesis (DNL). Furthermore, hyperglycemia will cause a compensatory increase in insulin secretion from pancreatic β cells to counter the insulin resistance [176]. High basal and continuously increasing fasting insulin level is a marker for the development of non-alcoholic fatty liver diseases (NAFLD) [177], and elevated insulin would increase SREBP-1c, coordinating LXR and ChREBP to promote the expression of lipogenic genes and de novo lipogenesis in the liver [50].

Furthermore, due to insulin resistance and elevated blood glucagon levels, there is an increase in adipose tissue lipolysis and the release of free fatty acids (FFA) into circulation in patients with T2DM and obesity [178]. Plasma FFA can be re-esterified and channeled into VLDL (Figure 5). Moreover, impaired insulin signaling can prevent the degradation of FOXO1 and phosphorylation of this protein by PKA can increase FOXO1 nuclear localization [35] and upregulate the expression of MTP [69]. Collectively, these effects will increase lipogenesis and VLDL production in the liver (Figure 5).

For lipogenesis in the liver, intact insulin signaling is required; therefore, defects in hepatic insulin signaling should lead to decreased fatty accumulation in the liver. However, patients with insulin resistance in obesity and T2DM commonly have hepatosteatosis [179,180]. Many patients with a similar degree of liver steatosis exhibit distinct insulin sensitivity (very high or very low) [181] and improvement of insulin resistance does not necessarily reverse liver steatosis [182]. Therefore, there is a dissociation of hyperglycemia and insulin resistance with liver steatosis in some patients. As mentioned above, endotoxemia can induce P300 via XBP1s in the ER stress pathway in the liver [67] and elevated P300 can disrupt insulin signaling by acetylating IRS1 and 2 in the insulin signaling pathway [67]. Nuclear form SREBP1 can activate SREBP1c gene expression in a feed-forward manner [47,183]. In the liver of HFD-fed mice, the nuclear form of the SREBP1 protein was heavily acetylated [139], and in cultured hepatocytes the mutation of P300 acetylation sites at K289 and K309 (double KR mutation) in SREBP1n (nuclear active form) completely abolished the binding of SREBP1n to the promoter of the SREBP1c gene. Furthermore, P300 can acetylate ChREBP at K672 to augment ChREBP-mediated lipogenesis (Figure 5) [184]. These studies revealed that endotoxin-induced P300 can cause insulin resistance and stimulate lipogenesis in the liver of obese and T2DM patients.

## 7. Perspective

Glucose is a major energy source for mammalian cells and takes a central position in metabolism. It is the sole energy source in some mammalian tissues and cells, such as neurons, erythrocytes, and the renal medulla; in the brain, the rate of glucose utilization is about 130 g/day [16,185]. However, higher blood glucose levels (fasting hyperglycemia) are toxic because of protein modifications and the induction of oxidative damage [186,187,188,189]. Hyperglycemia can cause other adverse effects in diabetes, including NAFLD [7,18,190]. Effective treatment to maintain the blood glucose levels of patients with obesity and T2DM within a small, defined range (70~110 mg/dL, fasting), therefore, is crucial in protecting patients from the adverse complications. 

Patients with T2DM and obesity commonly have NAFLD, which includes a complex spectrum of disorders ranging from steatosis to nonalcoholic steatohepatitis (NASH) and cirrhosis. Excessive accumulation of triglycerides (TG) is a hallmark of NAFLD. NAFLD is rising rapidly and is the leading indication for liver transplantation worldwide. NAFLD affects over 30% of the population in the USA [191]. NAFLD is now recognized as the liver disease component of metabolic syndrome because patients with NAFLD often have obesity and type 2 diabetes with insulin resistance, dyslipidemia, hypertriglyceridemia, and hypertension [192]. In patients with T2DM, over 75% of patients have NAFLD and over 90% of severely obese patients that underwent bariatric surgery have NAFLD [193,194]. NAFLD can lead to NASH, cirrhosis, and hepatocellular carcinoma. The majority of patients with NAFLD are also obese and/or have T2DM [2], suggesting that there may be a mutual cause-and-effect relationship between fatty liver and insulin resistance in a unified pathogenesis mechanism [195,196]. As such, hyperinsulinemia in obesity and diabetes results in the development of steatosis through increasing hepatic lipogenesis; subsequently, the accumulation of hepatic lipid species, e.g., diacylglycerol (DAG) and ceramide, leads to the impairment of insulin signaling [195,196,197]. For these reasons, several drugs, such as metformin, SGLT-2 inhibitors, TZD, and GLP-1 agonists, are used in the treatment of T2DM have been assessed for improving fat metabolism in patients with NAFLD.

However, as of now, effective treatment for NAFLD has not been found yet and needs to be developed. Since sedentary lifestyles are an important cause of obesity and T2DM, and exercise can improve insulin sensitivity and attenuate fat accumulation in the liver [198], increasing physical activities would be an effective intervention to combat obesity and T2DM.

## Figures and Tables

**Figure 1 life-13-00515-f001:**
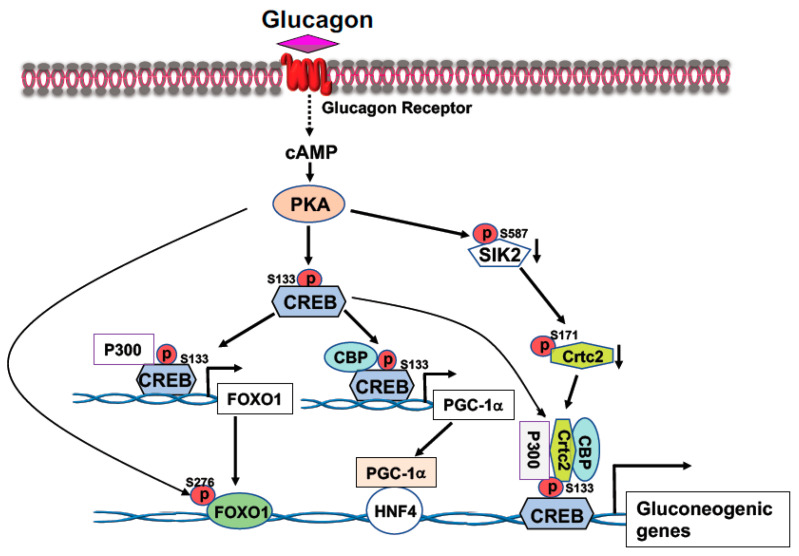
Activation of the cAMP-PKA signaling by glucagon stimulates gluconeogenic gene expression. Phosphorylation of CREB at S133 by PKA facilitates the bindings of CBP and P300 to CREB. Activated PKA also promotes Crtc2 binding to CREB through inhibition of SIK2 activity, leading to the formation of the CREB-CBP/P300-Crtc2 complex on the promoter of gluconeogenic genes. Phosphorylation of CREB at S133 by PKA recruits P300 on the promoter of *Foxo1* gene to initiate *Foxo1* gene expression, together with PKA-mediated FOXO1 phosphorylation at S276 increases its stability, leading to increased FOXO1 binding onto the promoter of gluconeogenic genes. Increased expression of *PGC-1a* gene expression by PKA-mediated CREB phosphorylation at S133 upregulates the rate-limiting gluconeogenic gene expression of *Pck1* and *G6pc* via HNF4a.

**Figure 2 life-13-00515-f002:**
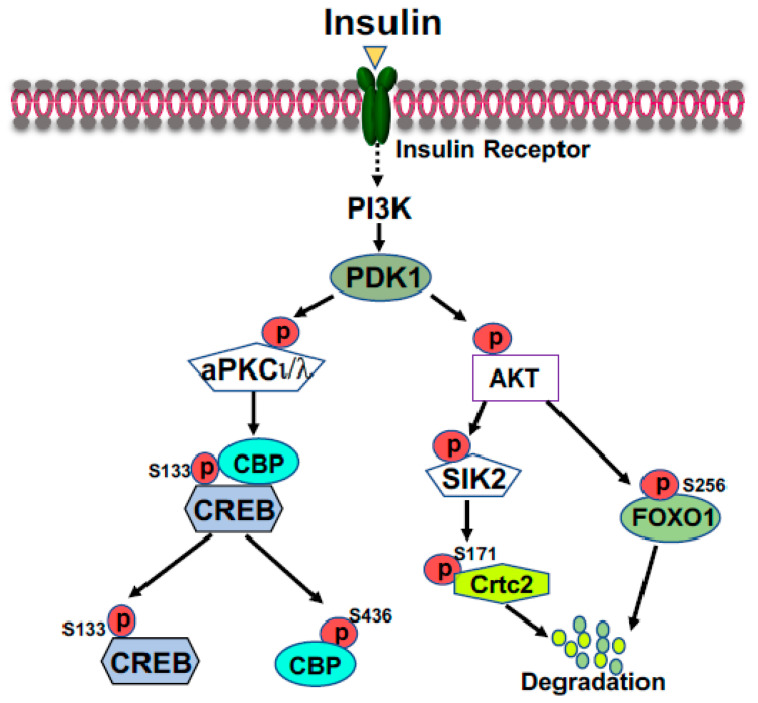
Suppression of gluconeogenic gene expression by insulin in the fed state. Phosphorylation of FOXO1 by insulin-activated AKT promotes FOXO1 nuclear exclusion and degradation. Activated AKT can activate SIK2, leading to the phosphorylation of CRTC2 at Ser171; subsequently, phosphorylated CRTC2 is excluded from the nucleus and degraded in the cytoplasm. Activated aPKC*ι*/*λ* by insulin, through the PI3K-PDK1 pathway, can directly phosphorylate CBP at S436, resulting in the disassembly of CREB coactivator complex.

**Figure 3 life-13-00515-f003:**
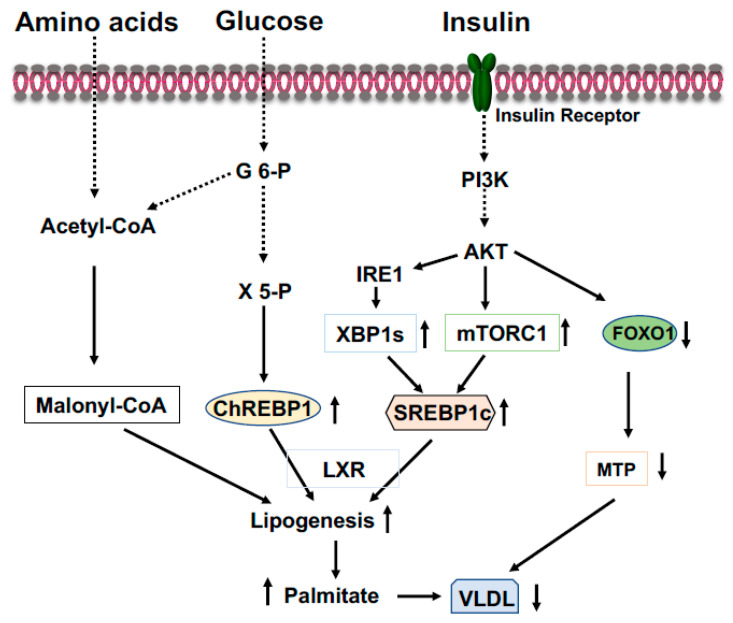
Coordinated regulation of lipid metabolism in the liver in the fed state. Elevated blood glucose and amino acids provide lipogenic substrate malonyl-CoA through conversion of acetyl-CoA. Activation of the PI3K-AKT signaling by insulin upregulates expression of SREBP1c through mTOCR1 and phosphorylation of IRE1 at S724 to mediate the splicing of XBP1 mRNA and the generation of spliced form XBP1s, then, XBP1s stimulates SREBP1c expression. Xylulose 5-phosphate via the pentose-phosphate pathway promotes ChREBP1 nuclear entry and lipogenesis. Phosphorylation of FOXO1 by AKT results in FOXO1 degradation and decreased expression of MTP, resulting in increased triglyceride accumulation in the liver.

**Figure 4 life-13-00515-f004:**
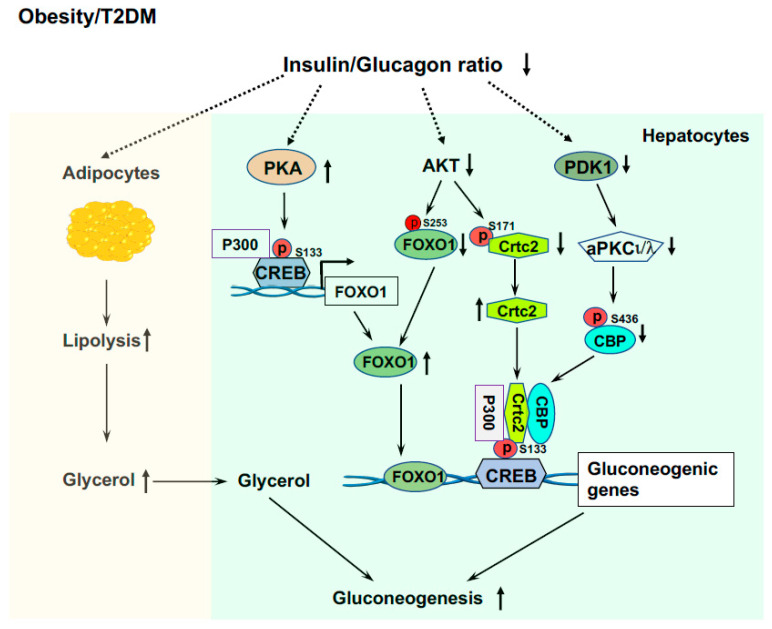
Elevated liver glucose production in T2DM and obesity. Elevated ratio of glucagon/insulin increases lipolysis in adipose tissue and released glycerol is a preferred gluconeogenic precursor. Increased P300 and CREB phosphorylation upregulate the expression of the *Foxo1* gene, and low AKT activity and impaired phosphorylation of FOXO1 by AKT, leading to increased FOXO1 protein levels that drive gluconeogenic gene expressions. Insulin resistance and impaired PDK1 and AKT activity leads to formation of the CREB-CBP/P300-Crtc2 complex to drive the over expression of gluconeogenic genes.

**Figure 5 life-13-00515-f005:**
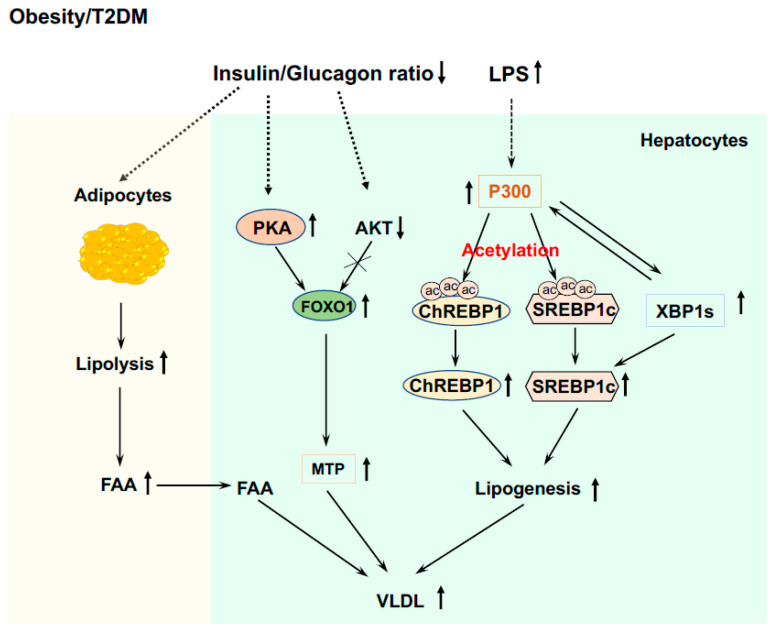
Abnormal lipid metabolism in the liver of T2DM and obesity. Induced P300 via XBP1s by endotoxemia acetylates SREBP1c and ChREBP1 to drive de novo lipogenesis. The interaction of P300 and XBP1s also stabilizes XBP1s, and XBP1s can stimulate de novo lipogenesis by upregulating SREBP1c expression. Impaired insulin signaling can prevent the degradation of FOXO1, and phosphorylation of FOXO1 by PKA can increase FOXO1 nuclear localization and upregulate the expression of MTP. The decreased ratio of insulin to glucagon stimulates lipolysis in adipose tissue and releases FFA into circulation, and FAA will be used to generate TG and incorporate into VLDL.

## Data Availability

Not applicable.

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
