# Peer review of "Regulation of Liver Glucose and Lipid Metabolism by Transcriptional Factors and Coactivators"

_life, 2023, doi:10.3390/life13020515_

Round 1

Reviewer 1 Report

In this review manuscript by Ramatchandirin et al, the authors delivered an overview of critical transcription factors and co-activators in hepatic glucose and lipid metabolism. This is more of an introductory type of article describing the key elements in the regulation of hepatic energy metabolism. While this manuscript is well written and provides useful outline of regulatory mechanisms, there are a number of concerns that need to be addressed before this manuscript is in a publishable fashion. Specific comments are as follows:  

1) The activity of transcription factors for hepatic glucose and lipid metabolism is enhanced by co-activators. The actions are negatively regulated by inhibitory phosphorylation and nuclear exclusion as described in the text. Are there known co-repressors that are worth to be included in this article?  

2) The nuclear receptors PPARs, LXRs, FXRs and HNF4 are crucial in hepatic metabolism but somehow are not in the whole picture of the coordinated regulation under normal or diseased conditions (Sections 4 to 6). It is recommended to at least describe how these factors are involved in normal (fasting and fed) glucose and lipid metabolism and possibly include them in the figures.  

3) CREB Regulated Transcription Coactivator 2 (Crtc2) was mentioned quite a few times but not introduced by its full name. With its importance in this article, this co-factor deserves its own subsection?  

4) Much of the information in Section 4 has been described in the previous sections and seems redundant. The authors may consider moving this section to the front, after the introduction.  

5) The proteolytic processing of SREBPs should be described, as the regulation by Insigs and full-length, nuclear forms of SREBPs are memtioned in other parts of the article.  

6) Minor: the terms such as XBP1u and SREBP1n should be defined. Line 351: the description "insulin can also phosphorylate CREB at S133" does not make sense. Please rephrase. 

Author Response

We would like to thank this reviewer for his/her comments concerning this review article.

1)The activity of transcription factors for hepatic glucose and lipid metabolism is enhanced by co-activators. The actions are negatively regulated by inhibitory phosphorylation and nuclear exclusion as described in the text. Are there known co-repressors that are worth to be included in this article?  

Response: We took reviewer’s suggestion and added a section of co-repressors regulation of metabolism (section 3.4).

2) The nuclear receptors PPARs, LXRs, FXRs and HNF4 are crucial in hepatic metabolism but somehow are not in the whole picture of the coordinated regulation under normal or diseased conditions (Sections 4 to 6). It is recommended to at least describe how these factors are involved in normal (fasting and fed) glucose and lipid metabolism and possibly include them in the figures.  

Response: These nuclear receptors and their roles in hepatic metabolism are discussed in the section 2.6, 4.1, and also included in the Figures 1, 3, 4.

3) CREB Regulated Transcription Coactivator 2 (Crtc2) was mentioned quite a few times but not introduced by its full name. With its importance in this article, this co-factor deserves its own subsection?  

Response: We added a new section on CRTC2 (section 3.2).

4) Much of the information in Section 4 has been described in the previous sections and seems redundant. The authors may consider moving this section to the front, after the introduction.  

Response: We first introduced separately about the important transcriptional factors and coactivators in the regulation of liver metabolism, then we want to give the pictures how these transcriptional factors and coactivators work together to modulate liver metabolism.

5) The proteolytic processing of SREBPs should be described, as the regulation by Insigs and full-length, nuclear forms of SREBPs are memtioned in other parts of the article.  

Response: The proteolytic processing of SREBPs was included in Section 2.4.

6) Minor: the terms such as XBP1u and SREBP1n should be defined. Line 351: the description "insulin can also phosphorylate CREB at S133" does not make sense. Please rephrase.  

Response: We defined XBP1u, the unspliced form, in Section 2.5.

SREBP1n, nuclear active form, was defined in Section 6.

A previous report showed that both insulin and glucagon can stimulate the phosphorylation of CREB at 133 (Fig. 1a, b, Koo, et al. Nature 2005, 437, 1109), suggesting that CREB is constitutively phosphorylated.

Reviewer 2 Report

There are far too many details uninteresting to the public, too generic, and unrelated to the main description. E.g., phosphorylation sites, nucleotide sequences, and so on.

Also, before discussing molecular aspects regulating glucose metabolism in the liver, the reader should be provided with an informative description of the main hormones and type of diet that initiate the molecular events presently envisaged. The lack of any mention of pyruvate dehydrogenase complex, which is the rate-limiting step in CHO oxidation is remarkable.

The effects of acute and chronic exercise on hepatic lipid composition on the regulation of the liver’s CHO and fat metabolism ought to be discussed.

There is far too much generic and repetitive information in the Introduction section.

The present reviewer would disagree with the authors' intuitive stance that various transcriptional factors and co-activators would be the primary players in the regulation of glucose and fat metabolism in the liver. The lifestyle (a diet rich in sugar and saturated fat, smoking, sedentary life, immobilization, medication, aging, and hormonal changes) would be the primary trigger of changes in liver glucose and fat metabolism.

Lack of description of key players in glucose metabolism like PDC and PDK1-4 kinases.

Lines 12-14 Simplify and correct the cause-effect relationship.

Lines 16-17 it does not make too much sense.

Lines 23-24. You start your time evolution with a percentage and end it with a number. A reader could not compare these two values.

Cardiovascular diseases are not metabolic abnormalities, but the former can be triggered by the latter.

Lines 51-53. The statement is quite incorrect as the skeletal muscles are the most important organ in regulating energy homeostasis. Compare 1.5 kg of the liver weight to that of 45 kg for muscle beds.

Line - 50 “Abnormal glucose metabolism with dysregulation impacts the major metabolic pathways, such as carbohydrates, and lipids, within different tissues and organs”. It does not make sense.

Lines 63-64 “Glucose is a principal energy fuel for mammalian cells and takes a central position in metabolism“. This statement is incorrect. This could occur only in the brain or during high-intensity exercise. Otherwise, fat is the primary energy fuel.

Lines 212-2134 – “PPARa is expressed mainly in the metabolic tissues, such as liver and skeletal muscle,

with high mitochondrial fatty acid oxidation.” The heart has the highest number of mitochondria of all organs.

The text requires English language proofing and factual confirmation.

Author Response

We would like to thank this reviewer for his/her comments concerning this review article.

1. There are far too many details uninteresting to the public, too generic, and unrelated to the main description. E.g., phosphorylation sites, nucleotide sequences, and so on.

Also, before discussing molecular aspects regulating glucose metabolism in the liver, the reader should be provided with an informative description of the main hormones and type of diet that initiate the molecular events presently envisaged. The lack of any mention of pyruvate dehydrogenase complex, which is the rate-limiting step in CHO oxidation is remarkable.

Response: We added a new paragraph in Section 4.1 to give a brief introduction of how to regulate glucose metabolism in the fed and fasted states by insulin and glucagon. We agree with reviewer that PDH plays important role in glucose metabolism in the mitochondria, however, this is out of the scope of current review.

2. The effects of acute and chronic exercise on hepatic lipid composition on the regulation of the liver’s CHO and fat metabolism ought to be discussed.

Response: We added a sentence in Perspective and  stated: "Since sedentary lifestyles are an important cause of obesity and T2DM and exercise can improve insulin sensitivity and attenuate fat accumulation in the liver [199], increasing physical activities would be an effective intervention to combat obesity and T2DM.  

3. There is far too much generic and repetitive information in the Introduction section.

Response: We would like to provide comprehensive information to the readers.

4. The present reviewer would disagree with the authors' intuitive stance that various transcriptional factors and co-activators would be the primary players in the regulation of glucose and fat metabolism in the liver. The lifestyle (a diet rich in sugar and saturated fat, smoking, sedentary life, immobilization, medication, aging, and hormonal changes) would be the primary trigger of changes in liver glucose and fat metabolism.

Response: We mainly discussed the molecular regulations of glucose and lipid metabolism. The lifestyles play critical roles in the development of obesity and T2DM through affecting insulin sensitivity and fat accumulation in the liver, etc.

5. Lack of description of key players in glucose metabolism like PDC and PDK1-4 kinases.

Response: As we pointed out in our above response that we agree with reviewer that PDH plays important role in glucose metabolism in the mitochondria, however, this is out of the scope of current review.

6. Lines 12-14 Simplify and correct the cause-effect relationship.

Response: We would like to present the facts about these diseases, not try to address the cause-effect relationship of T2DM and NAFLD.

7. Lines 16-17 it does not make too much sense.

Response: Carbon from pyruvate, lactate, amino acids can be converted into acetyl-CoA via PDH for lipogenesis, or converted into oxaloacetate via pyruvate carboxylase for gluconeogenesis. The carbon from these metabolites or amino acids can be used for either liver glucose and lipid metabolisms.

8. Lines 23-24. You start your time evolution with a percentage and end it with a number. A reader could not compare these two values.

Response: Both numbers will give the readers about the seriousness of these diseases. 

9. Cardiovascular diseases are not metabolic abnormalities, but the former can be triggered by the latter.

Response: We agree with reviewer and deleted the “cardiovascular diseases” in Line 31.

10. Lines 51-53. The statement is quite incorrect as the skeletal muscles are the most important organ in regulating energy homeostasis. Compare 1.5 kg of the liver weight to that of 45 kg for muscle beds.

Response: Liver is the largest metabolic organ (Campbell. Liver: metabolic functions. Anaesthesia & Intensive Care Medicine 2006, 7(2): 51). Skeletal muscle utilizes most of the glucose.

11. Line - 50 “Abnormal glucose metabolism with dysregulation impacts the major metabolic pathways, such as carbohydrates, and lipids, within different tissues and organs”. It does not make sense.

Response: It was our omission. We deleted “with dysregulation”.

12. Lines 63-64 “Glucose is a principal energy fuel for mammalian cells and takes a central position in metabolism“. This statement is incorrect. This could occur only in the brain or during high-intensity exercise. Otherwise, fat is the primary energy fuel.

Response: We rephrased the sentence and stated: “Glucose is one of the principal energy fuels for mammalian cells and takes a central position in metabolism.”

13. Lines 212-2134 – “PPARa is expressed mainly in the metabolic tissues, such as liver and skeletal muscle, with high mitochondrial fatty acid oxidation.” The heart has the highest number of mitochondria of all organs.

Response: We rephrased the sentence and stated: “PPARa is highly expressed in the metabolic tissues, such as liver and skeletal muscle, with high mitochondrial fatty acid oxidation.” 

14. The text requires English language proofing and factual confirmation.

Response: We carefully checked the grammar in this revision. 

Reviewer 3 Report

No major concerns. The Figure captions contains labels A-C or A-D . I suggest adding those labels in the Figures too.

Author Response

We would like to thank this reviewer for his/her comments concerning this review article.

No major concerns. The Figure captions contains labels A-C or A-D . I suggest adding those labels in the Figures too.

Response: To avoid the confusion, we removed the labels in figure legends.

Round 2

Reviewer 1 Report

The reviewer appreciates the authors for their feedback and revision of the manuscript. There are two additional minor questions:

1) In line 396, the description “Interestingly, insulin can also phosphorylate CREB at S133” is merely a grammatical question, as insulin does not directly “phosphorylate” CREB. Instead, insulin stimulates/increases phosphorylation of CREB.

2) In line 449, are “DNA proteins” DNA-binding proteins or do they have more specific terms?

Author Response

We appreciate reviewer's comments and corrected these mistakes.

Reviewer 2 Report

Modify as follows: "The liver is an the main organ that contributes to the regulation of blood glucose levels by releasing the glucose into circulation when blood glucose levels are low and stores along skeletal muscles excess of glucose as glycogen in post-prandial states."

Modify as follows: “In mammals, the liver is the central organ for fatty acid metabolism and a key player in glucose metabolism. The liver acts as a crossroad, which metabolically connects various organs, especially skeletal muscle, and adipose tissues, in the strive for maintaining the long-term energy supply to the body. The regulation of liver glucose metabolism and lipid metabolism is tightly controlled by dietary, hormonal, and neural signals through the activation of various transcriptional factors and co-activators”

As a way of exemplification, the authors themself cite that: "Skeletal muscle is crucial for glucose utilization and is responsible for over 80% of glucose clearance with the rest being used by the liver."

To make an additional point, after a standardised meal over 50% of the glucose ingested will be stored/oxidised by the muscle beds in a resting state. The brain and the liver will equally share 30% of the total glucose ingested.

Overall, the authors still appear to assign the liver the most important role in controlling whole-body glucose homeostasis. Therefore, the manuscript should constrain this personal and unjustified paradigm throughout the text.

Author Response

We appreciate reviewer's thoughtful comments and made changes as suggested by reviewer. We toned down the role of liver in metabolism and stated as: "The liver is one of the main organs that ---".